# Impacts of Air Velocity Treatments under Summer Conditions: Part II—Heavy Broiler’s Behavioral Response

**DOI:** 10.3390/ani12091050

**Published:** 2022-04-19

**Authors:** Suraiya Akter, Yingying Liu, Bin Cheng, John Classen, Edgar Oviedo, Dan Harris, Lingjuan Wang-Li

**Affiliations:** 1Department of Biological and Agricultural Engineering, North Carolina State University, Raleigh, NC 27695, USA; sakter@ncsu.edu (S.A.); lyy@njau.edu.cn (Y.L.); chengbin0228@gmail.com (B.C.); classen@ncsu.edu (J.C.); 2Department of Automation, College of Artificial Intelligence, Nanjing Agricultural University, Nanjing 210095, China; 3Prestage Poultry Science Department, North Carolina State University, Raleigh, NC 27695, USA; eooviedo@ncsu.edu; 4Department of Statistics, North Carolina State University, Raleigh, NC 27695, USA; doharris@ncsu.edu

**Keywords:** heavy broiler, heat stress, air velocity, behavior

## Abstract

**Simple Summary:**

Behavioral changes are one of the mechanisms for broilers to adjust their body temperature under heat stress conditions. However, the behavioral responses of heavy broilers to environmental changes have not yet been studied well. Therefore, this research investigated the behavioral changes of broilers under two dynamic air velocity treatments (high and low) under summer conditions. Video data collected from a heat stress experiment conducted on broilers aged 42–54 days were used to investigate variations in the number of chickens feeding, drinking, standing, walking, sitting, wing flapping, and leg stretching. The results indicated that the high air velocity treatments increased the number of chickens feeding, standing, and walking. In addition, age significantly affected the number of birds feeding, drinking, panting, and sitting, while the time of the day also affected the number of chickens drinking and panting. This study reveals the thermal stress of heavy broilers from their behavior under summer conditions to help manage the performance and welfare of birds under environmental stress.

**Abstract:**

Broiler chickens exposed to heat stress adapt to various behavioral changes to regulate their comfortable body temperature, which is critical to ensure their performance and welfare. Hence, assessing various behavioral responses in birds when they are subjected to environmental changes can be essential for assessing their welfare under heat-stressed conditions. This study aimed to evaluate the effect of two air velocity (AV) treatments on heavy broilers’ behavioral changes from 43 to 54 days under summer conditions. Two AV treatments (high and low) were applied in six poultry growth chambers with three chambers per treatment and 44 COBB broilers per chamber from 28 to 61 days in the summer of 2019. Three video cameras placed inside each chamber (2.44 m × 2.44 m × 2.44 m in dimension) were used to record the behavior of different undisturbed birds, such as feeding, drinking, resting, standing, walking, panting, etc. The results indicate that the number of chickens feeding, drinking, standing, walking, sitting, wing flapping, and leg stretching changed under AV treatments. High AV increased the number of chickens feeding, standing, and walking. Moreover, a two-way interaction with age and the time of day can affect drinking and panting. This study provides insights into heavy broilers’ behavioral changes under heat-stressed conditions and AV treatments, which will help guide management practices to improve birds’ performance and welfare under commercial conditions in the future.

## 1. Introduction

Broiler chickens are now bred to reach their market size weight of about 2.3 to 4.5 kg at the age of 42 to 63 days due to the demand for deboned meat compared to the whole bird [1,2]. Faster growth and heavier body weight in a confined facility often challenge these birds’ performance and welfare [3]. Moreover, global warming and climate change cause more heatwaves in the summertime. Consequently, birds are experiencing heat stress more often in summer due to increased temperature and relative humidity. Heat stress increases mortality rate [4,5] and feed conversion ratio [6,7], and decreases feed intake [7,8,9] and body weight gain [6,7,9]. This not only leads to economic loss but also compromises animal welfare.

Chickens adapt to heat stress by adopting several behavioral changes to maintain their homeostasis [10,11]. For example, birds eat less and drink more water [12,13]. Moreover, birds tend to sit, elevate their wings, and pant to dissipate excess heat produced from metabolism [7,14]. Therefore, when heat-stressed birds cannot release their body heat to the environment, they try to transfer heat to the environment through various activities and behavioral changes. In other words, these kinds of behavioral changes are indicators of their discomfort, and they provide further evidence of compromised welfare. Hence, it is essential to understand how birds respond under thermal stresses to provide them with the necessary support and means to ensure performance and welfare.

Several studies have been conducted to help understand broiler chickens’ behavioral changes under different environmental conditions and other management strategies, such as through the use of dietary manipulation or the addition of supplements [11,15,16]. Adding different levels of propolis in feed to heat-stressed broilers increased walking but did not change feeding, drinking, wing elevation, or preening in 15–42-day-old chickens [11]. However, synbiotic-fed 15–42-day-old broilers showed less panting and wing spreading and more standing, sitting, walking, feeding, and preening [16]. The increased light intensity significantly affected 35-day-old broilers in behaviors such as lying, eating, drinking, standing, walking, preening while lying, wing/leg stretching, sleeping, dozing, vocalization, and idling [17,18]. Since diet manipulation does not always impact broiler behavior, these approaches do not consistently reduce heat stress. Moreover, none of these approaches were investigated for current market-sized broilers.

Controlling the inside environment of broiler grow-out houses is being recommended by several researchers [14,19] to reduce heat stress impact on broilers. Various studies have been conducted to verify the impact of air velocity (AV) on the thermal comfort of broilers under stressful conditions [20,21,22,23]. Since producers are growing heavier birds, it is even more crucial to understand the behavioral responses of bigger birds to provide a comfortable environment. However, no researchers have yet studied birds’ behavioral changes pattern from 49 to 61 days under various air velocity treatments. Hence, the objective of this study was to investigate heavy broilers’ behavioral responses to AV treatments under heat stress conditions. The effect of AV on feeding, drinking, standing, walking, sitting, panting, wing flapping, and leg stretching was investigated. The variation in these behaviors according to time of day and the age of the broilers was also examined.

## 2. Materials and Methods

### 2.1. Experimental Unit

The experiment was conducted in the poultry engineering laboratory (PEL) of North Carolina State University (NCSU) in the summer of 2019 from 29 May to 1 July. The PEL has six simulated poultry chamber systems with core chambers with dimensions of 2.44 m × 2.44 m × 2.44 m for the birds’ stay (Figure 1). All these chambers are equipped with a nipple drinker line, four feeders, and an automated switch-timer soft lighting system. A belt-driven blower controlled by a variable frequency drive (VFD) system provides various ventilation rates and desired airspeeds to all the chambers in the range of 0.9–4.6 m/s at birds’ heights according to their age and the ambient temperature. More detailed descriptions of the PEL and its operations are reported by Wang-Li [24], West [25], and Shivkumar [26].

### 2.2. Animals

A total of 400 male COBB 500 broilers were hatched and raised in the floor pens under similar conditions at the NCSU poultry unit. Then, 264 birds without leg defects were randomly selected to be placed in the six experimental chambers, with 44 birds per chamber after 28 days. They remained in the chamber until reaching the age of 61 days. The final stocking density was ≤40 kg/m^2^, following the animal welfare guideline.

### 2.3. Core Chamber Environmental Data Monitoring

Each chamber’s air temperatures (Ta) were monitored with a thermocouple, and a HOBO Pro v2 External T/RH Data Logger, Model U23-002 (Onset, Computer Corporation, MA, USA), was placed at the airflow inlet and outlet of each chamber at the birds’ height. Calibrated thermocouples (range: −5 °C to 50 °C, and accuracy: ±0.002 °C) recorded temperature data continuously at 1 min intervals, while the HOBO logged both Ta and RH at 10 min intervals. The hourly average Ta values from both sensors were then averaged to obtain the hourly average temperature at the inlet.

### 2.4. Air Velocity Treatments

Two sets of dynamic AV treatments (high AV and low AV) were designed depending on inlet Ta and bird age. The AV treatment design criteria are detailed in [27]. As shown in Table 1, high and low AVs were designed for each of the six following growth condition classes: below optimum, around optimum, above optimum (moderate), above optimum (severe), above optimum (life-threatening), above optimum (warning). Changes in AV were achieved with programs written for VFD. It is important to note that, unlike previous broiler studies [21,22,28] using static AV treatments, this study implemented dynamic AV treatment levels based on the age of the birds and the air temperature, a procedure which is highly recommended and widely used in the broiler industry.

The AV treatments began on the birds aged 35 days after the birds moved into the chamber for a week. As shown in Table 1, when the Ta was below optimal temperature, there were no AV differences between the two treatments in order to avoid cold stress, which would have complicated the experiments. For broilers aged 35–61 days, high AV treatments were applied to chambers 1, 3, and 5 and low AV treatments to chambers 2, 4, and 6. The difference in AV varied depending on bird age and the extent to which the measured chamber Ta differed from its value under optimal thermal conditions.

### 2.5. Behavioral Data Collection

Three video cameras (DVR-4580, Swann Communications, Santa Fe Springs, CA, USA) were installed in each chamber on the left- and right-side walls and on the celling to capture videos of undisturbed bird activity. The video recordings were saved in an external hard drive connected to each camera. Unfortunately, the hard drive that stored all the recordings of the cameras in chamber 2 and the ceiling cameras of all six chambers was destroyed; hence, only the two side wall video recordings of five chambers were available. Segmented videos were selected based on video quality, availability from both side cameras for the same period, and undisturbed bird appearance, as in Figure 2. The available videos lasted 10–20 min for various ages (43, 44, 49, 51, and 54 days) and times of day (early morning, morning, noon, afternoon, evening, and night); 8–10 min spans of these videos were watched manually to count the number of chickens for various behavioral poses. The classification of the time of day was as follows: early morning: 5:00–8:00, morning: 8:00–11:00, noon: 11:00–13:00, afternoon: 13:00–17:00, evening: 17:00–20:00 and night: 20:00–24:00.

The number of chickens feeding, drinking, walking, and standing (up on their feet but not feeding, drinking, or walking), panting, resting, stretching legs, or wing flapping was counted manually by an observer for each individual video. The ethogram in Table 2 [11,29] was used for observing various behaviors.

No specific chickens were marked or colored for observation; hence, only the number of chickens engaging in any behavior listed in Table 2 was counted from the videos. Only one chicken from chambers 3 and 4 died on the 53rd day. Hence, the total number of chickens in each chamber was 44 for the video monitoring periods selected.

### 2.6. Statistical Analysis

The data were analyzed using Rstudio (version 1.0.143) (RStudio, Boston, MA, USA). The number of birds having different behavior was the average from the replicated chambers under the two treatment AVs. A two-way ANOVA test was used to analyze the effect of treatment, age, time of day, and interactions on the number of chickens engaging in various behaviors. The main effects and the interactions were considered significant at *p* < 0.05. If any factor had main effects, the Tukey HSD test was performed to check the differences at the level of that variable. The replicated chambers were considered blocking factors and the number of chickens was considered the experimental unit.

## 3. Results

### 3.1. Environmental Conditions

The hourly averages of Ta and RH varied by time of day (Figure 3). There was no significant difference in Ta and RH in high and low AV treatment chambers. The hourly Ta values for inlets at 49, 51, and 54 days were higher than for those at 43 and 44 days as the later days were warmer. The average hourly Ta during the video recording days was 24.84 ± 4.16 °C, while the average hourly RH was 68.37 ± 15.42%. Higher hourly Ta values and lower RH values were observed during the afternoon as compared with other times of day. In general, RH was higher in the early morning.

The experiment was designed to obtain the behavioral changes of broiler chickens due to AV treatments under heat stress conditions. Figure 4 represents the time distribution of the AV treatments under the different growth conditions defined in Table 1. The birds were under heat stress conditions when the Ta was in one of the four growth conditions (i.e., moderate, severe, life-threatening, and warning). The data presented in Figure 4 reflect only the five days of video recording. The inlet Ta exceeded the optimal growth condition 35% of the time during those days. The Ta was below or around its optimum value during the 43rd and 44th days. On the 49th day, AV was primarily moderate; however, 26% of the time, the condition was severe. The growth condition never reached the AV warning condition during the observation period. There was a 22% life-threatening growth condition on the 54th day, when there was also an 18% severe condition. During these five days, there were no occurrences of the warning condition. The severe and life-threatening conditions mainly occurred during the afternoon and evening.

### 3.2. Effect of the AV Treatments on Behavior

#### 3.2.1. Feeding

Treatment and age significantly affected the number of chickens engaging in feeding behavior (Table 3). The number of birds feeding decreased with age (Figure 5). The number of chickens feeding was significantly higher under high AV treatment (*p* < 0.05) than low AV treatment (Table 4). Although the time of the day did not affect the number of birds feeding, the interaction with the treatment affected the number of chickens (Table 3).

#### 3.2.2. Drinking

The number of chickens drinking did not vary under AV treatment (Table 3). Although the treatment had no main effect on the number of chickens drinking, the interaction with the time of day was significant (*p* < 0.05) (Table 3). The number of chickens drinking significantly changed according to time of day, age, and their interaction (Table 3).

#### 3.2.3. Standing and Walking

The numbers of chickens standing and walking were analyzed together, since it was difficult to distinguish between the two behaviors as there was no marking on the birds’ bodies. The numbers of birds standing and walking were affected by AV treatments and their interaction with the time of day (Table 3). More birds were standing and walking under high AV treatment than low AV treatment (Figure 5). The time of day did not have main effects, but its interaction with the time of day significantly changed the number of chickens standing or walking. Age had no effect on the number of birds standing or walking.

#### 3.2.4. Panting

Panting is one of the expected behaviors for maintaining thermoregulation under heat stress. The AV treatment had no main effect on the number of birds panting (Table 3). However, time of day, age, and their interaction significantly affected the number of panting birds. The number of birds panting increased with age (Figure 5). More birds were panting on day 49, 51, and 54 than on day 43 and 44 (Table 4). From noon to evening, more birds were panting than at night, in the early morning, or in the morning (Table 4).

#### 3.2.5. Sitting

The AV treatment significantly affected the number of chickens sitting (Table 3). More birds were sitting under low AV treatment (Table 4). Age had main effects on the number of birds sitting (Table 3). More chickens were resting on day 44 than on day 49. Time of day had no main effect on the number of chickens sitting, but the interaction with AV treatment significantly affected the number of birds sitting. The interaction between age and AV treatment also changed the number of birds sitting.

#### 3.2.6. Wing Flapping and Leg Stretching

Wing flapping and leg stretching are two common behaviors under heat stress. This study discovered that the AV treatment significantly changed the number of birds flapping their wings and stretching their legs (Table 3). The number of birds flapping their wings or stretching their legs was higher under high AV (Figure 5). Age and the time of day did not have any effect on these behaviors, but the interaction between AV and age significantly changed the number of birds flapping their wings or stretching their legs under heat stress conditions.

### 3.3. Bird Sitting Location

The chickens moved around the chamber for various purposes. For example, they moved to the feeder location for feeding and moved to the middle of the chamber for drinking. While resting, they tended to sit at various locations. The number of chickens sitting varied at the inlet and the outlet (Table 5). Under both AV treatments, the number of chickens sitting near the inlets was significantly higher (*p* < 0.05) than the number of those sitting near the outlets for chickens of any age and at any time of day (Table 5 and Figure 6). Time of day and age did not impact the number of chickens sitting at the inlet or the outlet.

## 4. Discussion

Under heat stress, birds usually reduce their feed intake according to their age up to 42 days [9,10]. Although the time budget for feeding and feed intake was not determined in this study, the decreased number of chickens feeding implies that decreased feeding occurs at later ages (42–54 days). A heavier body weight at a later age lessened birds’ activity under heat stress. As a result, they could not release enough metabolic heat to bring their body temperature to the thermoneutral zone. Hence, their coping mechanism was to eat less in this context. The high AV treatment helped more chickens to eat under heat-stressed conditions. Hence, increased AV and proper mixing of AV can help regulate broiler performance even under faster growth rates and environmental stresses.

Irrespective of AV treatment, drinking declined with broiler age. Bizeray [10] and Jacobs [30] found increased drinking behavior in broilers up to six weeks of age, while Newberry [31] found a decreased water intake from the sixth to the ninth week under summer conditions. Newberry [31] found no time-of-day effect on drinking behavior, since the Ta was controlled and remained the same throughout the experiment. In this study, the AV treatments were dynamically changed with changes in Ta and bird age. Hence, it was reasonable to expect variations in the number of chickens drinking water according to time of day. Birds often drink more under heat stress [32], but due to their heavier body weight in this study, the birds tended to walk less, which might be another reason for the decrease in the number of chickens drinking. An interesting but not statistically significant observation is that during the early morning on any day, at noon and in the morning on the 49th day, and in the evening of the 51st day and the night at 54th day, more birds were drinking under high AV treatment (Figure 4). The collected videos from the low AV chambers were all during the lights-off condition for the early morning. Hence, the birds had almost zero activity in those chambers, leading to fewer chickens drinking. The same condition is applicable for the 54th day’s night observation. On the 49th day, the AVs in both treatments were from moderate to severe all day long, except for the early morning. Hence, under both treatments, the birds tended to drink more.

The birds selected for this experiment were all without leg defects. During the experimental period, the birds’ activity was more involved in either feeding or drinking. The heavier broilers were found to be less likely to walk or stand. It was even observed that they barely made more than two or three steps unless it was required to reach the drinker or feeder. The heavier body weight and stressful weather condition made them sit more often than walk or stand. Li [33] found restless walking behavior under heat stress in 21-day-old birds, but during this experiment, less walking and standing were observed among birds aged 42–54 days. As the birds aged, they sometimes resorted immediately to standing. Moreover, lameness or laziness was prominent when the thermal environment exceeded severe conditions. Less walking and standing indicate heat stress behavior for heavier birds. Since the number of chickens standing and walking increased under high AV treatment, increasing AV might be a good management strategy under heat-stressed conditions to cool down the birds and make them comfortable.

The absence of sweat glands and the presence of feathers increase broilers’ panting under higher environmental temperature to release excessive body heat. The AV treatments in this study did not significantly impact panting behavior. Hence, the number of chickens panting under the two treatments did not differ significantly. Panting increased with age (Figure 4). On the 49th, 51st, and 54th days, the number of chickens panting was higher than on the 43rd and 44th days. Under both treatments, the growth condition exceeded the optimum condition 72%, 46%, and 57% of the time on days 49, 51, and 54, respectively (Figure 3). Hence, the birds panted more often on those days. The number of birds panting was higher during noon, afternoon, and evening while the environment was warmer, which is consitent with the results of Lott et al. [34], who found that 4-to-6-week-old birds panted more often during warmer periods of the day while sitting on the floor pen at 0.25 m/s AV. A lower number of panting birds was observed in the early morning and morning. The growth condition was around or below optimum in the early morning. In the morning, the condition was mainly optimum or moderate, and the birds did not experience significant heat stress during those times; there was therefore less of a necessity to pant to release heat. At a later age, even in the nighttime, many birds were panting. This occurred because the growth condition reached severe and moderate on those days. Lott et al. [34] found decreased panting in birds aged 4 to 6 weeks under tunnel ventilation with AV in 2.08 m/s. Hence, the effect of AV requires further investigation to help heavy broilers manage under stressful conditions.

This study suggests that the number of birds sitting increases with age. Li [33] found that the duration of lying down increased with age up to 21 days in order to decrease basal metabolic rate and resist heat stress. Although our study did not analyze the duration of lying down or sitting, the higher number of birds indicates that they were more likely to sit at a later age. At an older age, the birds’ bodies were heavier, and hence they were less likely to walk or stand. Even activities for releasing heat, such as panting, wing flapping, or leg stretching, mostly occurred while the birds were sitting. Feeding and drinking activity also decreased with age, so that the number of birds sitting was higher under every treatment. Fewer birds were sitting under high AV treatment because high AV helped them release some heat and thereby feel comfortable enough to move for feeding, drinking, or other activities. Although the time of day did not impact the number of birds resting, its interaction with AV treatment impacted the number. Tao and Xin [35] suggest that broiler chickens’ core body temperature responds to the cumulative action of dry-bulb temperature, dew point temperature, and air velocity. Hence, in this experiment, the birds’ core body temperature also changed at different times of the day under different temperatures and corresponding air velocities, which led to changes in the number of birds sitting or resting.

Although the number of chickens engaging in wing flapping or leg stretching was small, the occurrences were observed under stressed conditions. On average, only a couple of chickens flapped their wings or stretched their legs, but at a later age, when the Ta was higher, at most 11 birds in the low AV treatment chamber were found engaging in this activity. They might have been trying to cope with the stressful environment with this behavior.

According to the ANOVA test, the blocking factor chamber had a significant impact on the number of chickens feeding, drinking, and panting (Table 3). Although the replicated chambers under high AV treatments were not significantly different according to environmental conditions (Figure 3), surprisingly, the number of chickens feeding, drinking, and panting was significantly higher in chambers 1 and 3 than in chamber 5. On days 43, 44, and 49, early morning and nighttime videos from chamber 5 were taken while the lights were off. Therefore, the number of chickens with different activities was very low at those times, which might have impacted the results. Hence, a balanced design and careful data collection processes are recommended for future investigation.

The birds moved around the chamber floor primarily to drink or eat. They sometimes walked simply to find a better location in the chamber. During warmer periods, the birds tended to sit more often near the inlet to experience a higher speed of air flow passing over their bodies. Since the air entered the chamber through the inlet and the air speed was higher there, the birds under heat stress wanted to sit more often at the inlet side compared to the outlet, where upwind birds may block some flow to reduce the air speed in the downwind zone. From this study, it was found that the heavier birds tended to stay closer to the location where air entered when conditions were warmer or stressful. Hence, it is important to make sure that all the birds in a facility can experience uniform air velocity passing over them to keep them comfortable. Bringing down air flow to birds’ height and properly mixing AV throughout the housing facility could be a good strategy for pacifying birds under heat stress. A collective understanding of these behavioral changes will help in understanding the potential issues associated with broilers’ welfare in the flock in order to take necessary management actions to maintain and/or enhance performance and welfare.

## 5. Conclusions

These results indicate that high AV treatment significantly changed the number of chickens engaging in feeding, standing, walking, sitting, wing flapping, and leg stretching behavior under heat stress. The numbers of chickens feeding and panting increased with age, but the number of those drinking and sitting declined. The number of chickens drinking and panting varied significantly according to the time of the day. The applied AV did not directly affect drinking and panting, but its two-way interaction with age and time of day significantly altered the prevalence of these behaviors under thermal stress. In general, heavy broilers changed their typical behaviors, such as feeding, drinking, walking, or resting, under heat-stressed conditions in order to adapt to the stressor. Air velocity can enhance the heat release activity of larger and older broiler chickens, ensuring their growth performance and welfare. The findings from this study will help to identify thermal stress through behavioral changes in the birds themselves. This will help producers make necessary management decisions to keep their birds healthy and happy under conditions of environmental stress.

## Figures and Tables

**Figure 1 animals-12-01050-f001:**
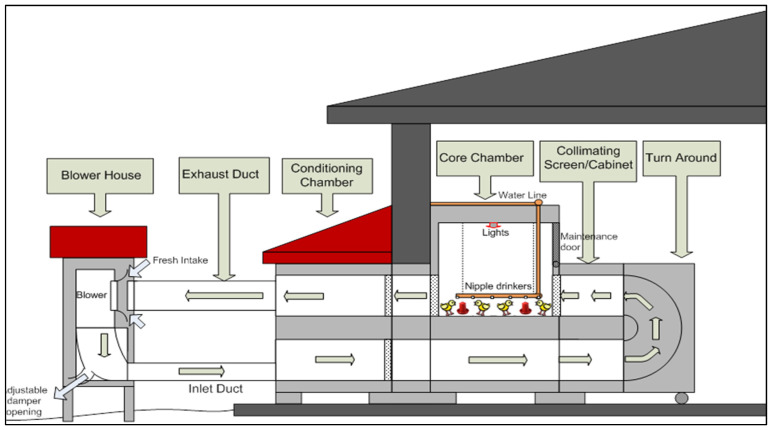
Cross section of the poultry chamber systems (from Shivkumar [26], used with permission).

**Figure 2 animals-12-01050-f002:**
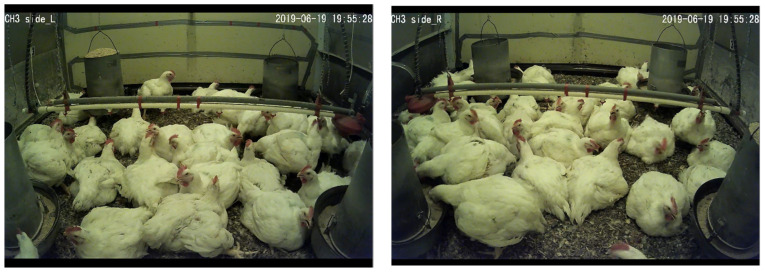
Snap shots of videos from the same time from both (**left**) and (**right**) cameras in a chamber, showing 2 chickens drinking, 4 feeding, 12 panting, 1 leg stretching, 0 wing flapping, 1 standing, and 24 sitting.

**Figure 3 animals-12-01050-f003:**
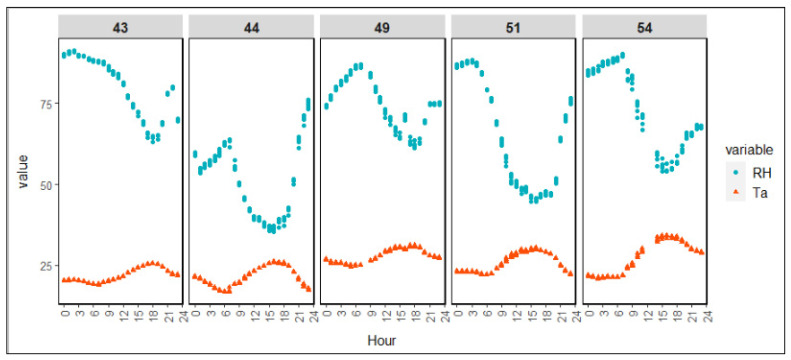
Average hourly Ta values and RH values by treatment during video observation periods.

**Figure 4 animals-12-01050-f004:**
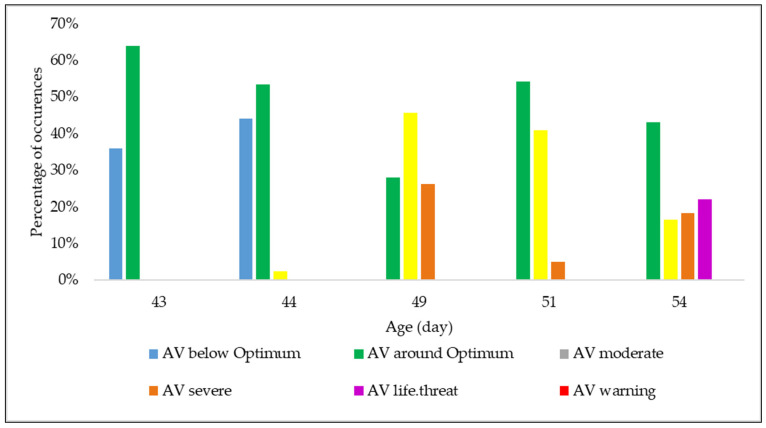
Time distribution of AV treatments implemented in all chambers only during video capture times.

**Figure 5 animals-12-01050-f005:**
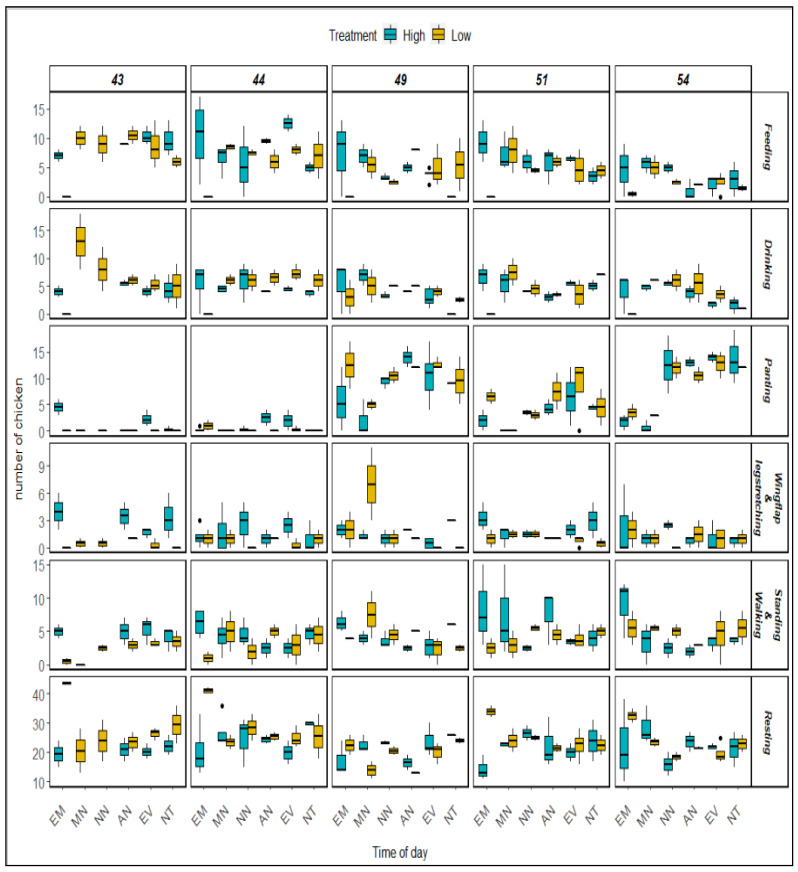
Broilers’ various behaviors separated by age and time of day (EM = early morning; MN = morning; NN = noon; AN = afternoon; EV = evening; NT = night) under high and low AV treatments.

**Figure 6 animals-12-01050-f006:**
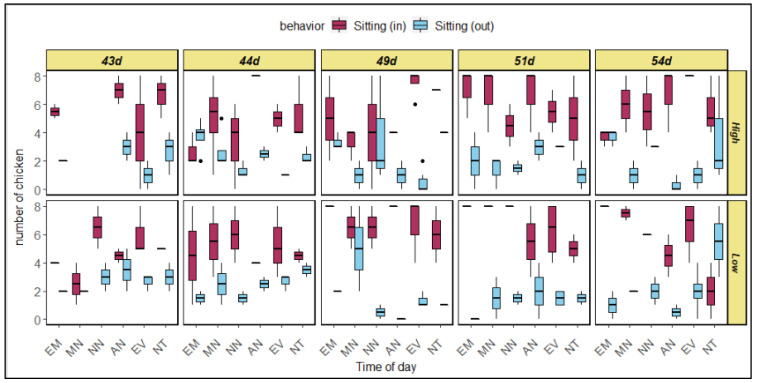
Difference in the number of chickens sitting at the inlets and the outlets of the chambers under both AV treatments during different times of the day (EM = early morning; MN = morning; NN = noon; AN = afternoon; EV = evening; NT = night.

**Table 1 animals-12-01050-t001:** High and low AV treatment design.

Treatment	Broiler Age (Days)	Temp °C	AV (m/s) Below Optimum T	Temp °C	AV (m/s) around Optimum T	Temp °C	AV (m/s) above Optimum T (Moderate)	Temp °C	AV (m/s) above Optimum T (Severe)	Temp °C	AV (m/s) above Optimum T (Life-Threatening)	Temp °C	AV (m/s) above Optimum T (Warning)
High	28–34 *	<26.0	0.9	26.0–27.8	1.23	27.8–28.9	1.33	28.9–32.2	1.48	32.2–33.9	1.64	>33.9	1.75
Low			0.9		1.23		1.33		1.48		1.64		1.75
High	35–40	<21.7	0.9	21.7–26.0	1.23	26.0–30.0	2.02	30.0–33.0	2.77	33.3–37.8	3.45	>37.8	3.95
Low			0.9		1.23		1.48		2.02		2.77		3.45
High	41–42	<21.1	1.48	21.1–26.0	1.48	26.0–30.0	2.02	30.0–33.0	2.77	33.3–37.8	3.45	>37.8	3.95
Low			1.48		1.48		1.48		2.02		2.77		3.45
High	43–52	<20.6	1.48	20.6–26.0	1.75	26.0–30.0	2.02	30.0–33.0	2.77	33.3–37.2	3.95	>37.8	4.33
Low			1.48		1.75		1.75		2.43		3.02		3.65
High	53–54	<19.4	1.48	19.4–25.0	1.75	25.0–29.5	2.43	29.4–32.7	3.02	32.7–36.1	3.95	>36.1	4.33
Low			1.48		1.48		1.75		2.43		3.02		3.65
High	55–56	<19.4	1.48	19.4–25.0	1.75	25.0–29.5	2.43	29.4–32.7	3.02	32.7–35.6	3.95	>35.6	4.33
Low			1.48		1.48		1.75		2.43		3.02		3.65
High	57–58	<18.9	1.48	18.9–25.0	1.75	25.0–29.5	2.43	29.9–32.2	3.02	32.2–35.6	3.95	>35.6	4.33
Low			1.48		1.48		1.75		2.43		3.02		3.65
High	59–60	<18.9	1.48	18.9–24.4	2.43	24.4–28.9	3.02	28.9–31.7	3.45	31.7–35.0	4.33	>35.0	4.43
Low			1.48		1.75		2.43		2.77		3.65		3.8
High	61	<18.3	1.48	18.3–23.9	2.43	23.9–28.9	3.02	28.9–31.7	3.45	31.1–33.9	4.33	>33.9	4.6
Low			1.48		1.75		2.43		2.77		3.65		3.95

* A non-treatment period (in the first week) allowed the broilers to acclimate to their new environment.

**Table 2 animals-12-01050-t002:** Bird behavior ethogram.

Behavior	Definition
Feeding	The bird’s head is located inside the feeder.
Drinking	The bird’s beak is in contact with the drinker.
Panting	The bird is breathing hard and quickly with a wide-open mouth and constantly shallow respiration.
Standing or Walking	Both feet are in contact with the floor; no other body part is in contact with floor.
Walking	The bird is in the process of taking at least 2 steps, including scratching the litter.
Sitting	Most of the ventral region of the bird’s body is in contact with the floor. No space is visible between the floor and the bird.
Wing flap	Flapping wings so that space can be seen between the bird’s wings and its body.
Leg stretching	Stretching one leg, often together with the wing of the same side, but the leg may also be stretched alone while sitting or standing.
The behaviors were mutually exclusive.	

**Table 3 animals-12-01050-t003:** Results of ANOVA test for differences in behavior according to AV treatment, age, and time of day.

Behavior	Effects (*p*-Value)						
	AV Treatment	Time of Day	Age	AV Treatment × Time of Day	AV Treatment × Age	Time of Day × Age	Chamber
Feeding	0.0171 *	0.1145	1.33 × 10^−9^ ***	1.10 × 10^−5^ ***	0.8334	0.4446	0.0205 *
Drinking	0.26698	4.69 × 10^−5^ ***	0.00353 **	2.26 × 10^−6^ ***	0.65882	0.01318 *	1.03 × 10^−5^ ***
Panting	0.593863	4.46 × 10^−13^ ***	<2 × 10^−16^ ***	0.285085	0.068111	2.30 × 10^−6^ ***	0.000473 ***
Standing and Walking	0.0478 *	0.1784	0.1844	0.0380 *	0.0175 *	0.6045	0.1844
Sitting	0.01777 *	0.14526	0.00945 **	1.38 × 10^−7^ ***	0.03087 *	0.27508	0.15007
Wing flapping and leg stretching	0.00493 **	0.2688	0.75742	0.17656	0.01473 *	0.68147	0.37366
df	1	5	4	20	4	20	4

Different asterisks represent different levels of significance (*** *p* < 0.001; ** *p* < 0.01; * *p*< 0.05, *p* < 0.1).

**Table 4 animals-12-01050-t004:** Differences in the average number of chickens with different behavior according to various factors.

Behavior	Number of Chickens (Mean ± Std)												
	AV		Age (Days)					Time of Day *					
	High	Low	43	44	49	51	54	EM	MN	NN	AN	EN	NT
Feeding	6.1 ± 3.8 a	4.9 ± 3.0 b	8.2 ± 3.5 a	7.3 ± 4.1 ab	4.5 ± 3.3 cd	5.6 ± 3.2 bc	2.9 ± 2.4 d	4.8 ± 3.4 a	6.9 ± 2.6 a	5.1 ± 3.1 a	5.9 ± 3.4 a	5.8 ± 3.6 a	4.9 ± 3.5 a
Drinking	4.2 ± 2.2 a	4.6 ± 3.3 a	5.4 ± 3.9 a	5.1 ± 2.5 a	4.0 ± 2.4 ab	4.6 ± 2.5 ab	3.5 ± 2.4 b	3.4 ± 3.5 c	6.3 ± 3.2 a	5.3 ± 2.5 ab	4.5 ± 1.8 abc	4.0 ± 1.9 bc	3.7 ± 2.4 c
Panting	4.8 ± 5.5 a	5.1 ± 5.5 a	0.7 ± 1.6 c	0.5 ± 1.1 c	9.2 ± 4.7 a	4.3 ± 4.0 b	9.3 ± 5.7 a	3.4 ± 4.2 c	1.1 ± 1.9 d	5.6 ± 5.5 abc	6.3 ± 5.7 ab	7.6 ± 5.9 a	5.1 ± 5.9 bc
Standing and Walking	5.6 ± 2.9 a	3.7 ± 2.3 b	3.3 ± 2.1 a	1.0 ± 2.4 a	3.9 ± 2.1 a	5.1 ± 3.5 a	4.4 ± 2.8 a	5.3 ± 3.7 a	4.5 ± 3.6 a	3.7 ± 1.8 a	4.1 ± 2.5 a	3.5 ± 1.9 a	4.2 ± 1.8 a
Sitting	22.4 ± 5.9 b	24.5 ± 7.0 a	24.8 ± 7.9 ab	25.9 ± 6.9 a	20.8 ± 4.5 b	22.9 ± 5.9 ab	22.7 ± 6.3 ab	25.2 ± 11.1 a	23.5 ± 5.7 a	23.1 ± 4.6 a	21.9 ± 4.7 a	22.1 ± 3.9 a	24.7 ± 5.4 a
Wing Flapping and Leg Stretching	1.7 ± 1.6 a	0.9 ± 1.6 b	1.5 ± 1.9 a	1.3 ± 1.5 a	1.5 ± 2.1 a	1.5 ± 1.3 a	1.2 ± 1.6 a	1.9 ± 2.0 a	1.7 ± 2.4 a	1.3 ± 1.4 a	1.3 ± 1.1 a	87 ± 1.1 a	1.3 ± 1.3 a

a–d Different letters within a row under the same factor mean a significant difference at level 0.05; * EM = early morning; MN = morning; NN = noon; AN = afternoon; EV = evening; NT = night.

**Table 5 animals-12-01050-t005:** Results of ANOVA test for the effect of testing location on broilers’ sitting behavior.

AV	Factors	Degrees of Freedom	Type III Sum of Squares	Mean Square	F-Value	*p* > F
High	Location (inlet/outlet)	1	441.6	441.6	104.587	<2 × 10 ^−0.5^ ***
	Time of Day	5	21	4.2	0.994	0.424
	Age	4	4.5	1.1	0.267	0.899
Low	Location (inlet/outlet)	1	473.5	473.5	127.459	<2 × 10 ^−0.5^ ***
	Time of Day	5	9.8	2	0.527	0.755
	Age	4	5	1.2	0.333	0.855

*** *p* < 0.001.

## Data Availability

The data presented in this study are available upon request from the corresponding author.

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
