# Peer review of "Impacts of Air Velocity Treatments under Summer Conditions: Part II—Heavy Broiler’s Behavioral Response"

_animals, 2022, doi:10.3390/ani12091050_

Round 1
Reviewer 1 Report
This manuscript explored the effects of different air velocity treatments on the behavior of broilers under heat stress. The experimental design was novel and the research method was feasible. The results can provide reference for the management of large-scale broiler production and animal welfare. However, there are still some problems that need to be carefully revised by the authors.
- Whether the cobb broilers raised in this experiment were under heat stress, or what was the specific threshold temperature or humidity of heat stress. These were the key information that needs to be added to the manuscript.
- Other minor errors were on line 17 (supplement “day”), line 18 (remove “chickens”), lin203 (title case “table”) and so on.
- All “d” in the manuscript should be changed to “day”.
- Revised the name of variable “T” in Figure 3.
Author Response
Respected Reviewer
Thank you for your comments and suggestions to improve this paper. Responses to each comment (in bold) are provided below. We hope the edits to the specific comments below aid in understanding
Sincerely,
Suraiya
- Whether the cobb broilers raised in this experiment were under heat stress, or what was the specific threshold temperature or humidity of heat stress. These were the key information that needs to be added to the manuscript.
The heat stress conditions in this experiment were the dynamic ambient conditions. The threshold temperature value for optimum environmental conditions varied over the age of the birds. Table 1 listed the temperature and corresponding classes that describe the stress condition inside the chamber was as “optimum, “moderate”, “severe” “life-threatening”, “warning”. The observed temperature and relative humidity values from the chamber were given in Figure 3. Also, Figure 4 explained how much time the birds were under heat stress condition i.e. Ta exceeded “optimum” growth condition.
- Other minor errors were on line 17 (supplement “day”), line 18 (remove “chickens”), lin203 (title case “table”) and so on.
Lines edited as per the recommendation.
- All “d” in the manuscript should be changed to “day”.
Corrected as per instruction
- Revised the name of variable “T” in Figure 3.
Revised and corrected
Reviewer 2 Report
Review Animals- -1668790
Title: Impacts of Air Velocity Treatments under Summer Conditions: 2 Part II—Heavy Broiler's Behavioral Response
The manuscript aims to investigate heavy broiler's behavioral responses to air velocities treatments under heat stress conditions. I believe the topic is relevant, the manuscript is well organized, and the results are well presented.
May I suggest including the paper by Tao, Xiangyi & Xin, Hongwei. (2003). Acute synergistic effects of air temperature, humidity, and velocity on homeostasis of market-size broilers. Transactions of the ASAE. 46. 10.13031/2013.12971. I believe it will be of great importance for your discussion.
Other particular suggestions and corrections are in the attached file.

Author Response
Respected Reviewer
Thank you for your comments and suggestions to improve this paper. Responses to each comment are provided accordingly in the manuscript. The suggested paper was also included as a relevant discussion point.
We hope the edits aid in the specific comments in understanding.
Sincerely,
Suraiya